# Event-Triggered-Based Fuzzy Control for Networked Control Systems with Compensation Mechanism against DoS Attacks

1st Yingnan Pan
*College of Control Science and Engineering*
*Bohai University*
Jinzhou, China
panyingnan@qymail.bhu.edu.cn

2nd Changhao Li
*College of Control Science and Engineering*
*Bohai University*
Jinzhou, China
lichanghao0128@163.com

*Abstract*—This paper investigates the design problem of event-triggered (ET) fuzzy controller for networked control systems subject to denial-of-service (DoS) attacks via compensation scheme. Firstly, for constrained communication bandwidth, an ET scheme is introduced to save network resources. Secondly, based on the DoS attacks compensation scheme, the unfavorable impact of DoS attacks on system is alleviated. Compared with most of the existing methods, the attack compensation mechanism introduced in this paper can maintain data transmission during DoS attacks. Combining the above content, the Lyapunov stability theory is utilized to identify the sufficient conditions for the closed-loop system to be asymptotically stable. Finally, simulation verification is presented to demonstrate the effectiveness of the proposed control strategy.

*Index Terms*—Event-triggered fuzzy controller, DoS attacks, compensation mechanism, networked control systems.

## I. INTRODUCTION

Nonlinear networked control systems (NCSs) have received extensive attention from the researchers with the development of control techniques [1, 2]. The Takagi-Sugeno (T-S) fuzzy model, as a model that rely on fuzzy rules to approximate nonlinear terms, has gradually entered scholars field of vision with the research on nonlinear NCSs [3–5]. For instance, the authors in [5] resolved the guaranteed cost static output feedback control problem for T-S fuzzy system by using guaranteed cost performance function. It should be pointed out that the above literature did not take into account the challenge of constrained communication bandwidth in NCSs.

In the NCSs, due to the limited communication bandwidth, the reduction of the occupation on network resources for NCSs has become a concern for many researchers. To save limited communication resources, event-triggered (ET) mechanisms have been proposed and widely applied [6, 7]. To name a few, in [7], the authors investigated the dynamic output feedback model predictive control problem for NCSs with packet loss and bounded disturbance under the fixed threshold ET scheme. In contrast to time-triggered mechanisms, the ET scheme mitigates redundant packet transmissions and selectively forwards

This work was partially supported by the Revitalization of Liaoning Talents Program (XLYC2203201).

packets that meet ET criteria to the network. It should be noted that the above results only considered the issue of saving communication resources and did not take into account the possibility of malicious attacks on communication networks.

A typical form of network attacks is denial-of service (DoS) attacks. When DoS attacks are conducted on the communication network by attackers, the transmission of data will be blocked, which severely impacts the system performance. To reduce the unfavorable impact of DoS attacks on the system, many remarkable works have been published [8, 9]. The authors in [8] proposed a security-based ET fuzzy control method for NCSs to reduce the unfavorable impact of DoS attacks. Fan *et.al* [9] designed the resilient sampled-data controller for DoS attacks, reduced the effect from DoS attacks by adjusting the sampling instant. Recently, researchers mainly deal with DoS attacks in a resilient mechanism by increasing data transmission instead of considering the transmission issues when DoS attacks happen. Therefore, investigating a DoS attacks compensation mechanism to maintain the transmission when DoS attacks happen is a significant and challenging issue.

Taking inspiration from the above, this paper investigated the problem of designing an attack compensation controller based on ET mechanism which mitigates the unfavorable impact of DoS attacks while saving communication resources. In this paper, an ET scheme is introduced to save network resources and then a DoS attack compensation mechanism is applied in the ET to controller channel to maintain the transmission of the data when DoS attacks happen, thereby achieving the purpose of weakening unfavorable effects of DoS attacks. Considering ET mechanism and DoS attacks, a controller design method is proposed for T-S fuzzy NCSs to ensure asymptotic stability and security of the system.

## II. PROBLEM FORMULATION

### A. T-S Fuzzy Model

Consider the T-S fuzzy model with $\hat{n}$ rules, giving the description as follows:

**Plant Rule** $l$: **IF** $\eta_1(x(k))$ is $\mathcal{V}_1^l$, and $\eta_2(x(k))$ is $\mathcal{V}_2^l$ and ,..., and $\eta_p(x(k))$ is $\mathcal{V}_p^l$, **THEN**

$$x(k+1) = \mathbb{A}_l x(k) + \mathbb{B}_l u(k) + \mathbb{B}_{1l} w(k),$$
$$z(k) = \mathbb{C}_l x(k) + \mathbb{D}_l u(k), \tag{1}$$

where $\mathcal{V}_\alpha^l$ is the fuzzy set ($l = 1, \ldots, \hat{n}$ and $\alpha = 1, \ldots, \text{p}$). Define $\eta_\alpha(x(k)) = \eta_\alpha(x)$ to be premise variable. $p$ denotes the number of the premise variables. $x(k) \in \mathbb{R}^x$ is the state vector. $u(k) \in \mathbb{R}^u$ is the input vector. $\omega(k) \in \mathbb{R}^\omega$ is the external disturbance that belongs to $\ell\,[0, \infty)$. $z(k) \in \mathbb{R}^z$ is the controlled output. $\mathbb{A}_l$, $\mathbb{B}_l$, $\mathbb{B}_{1l}$, $\mathbb{C}_l$ and $\mathbb{D}_l$ are known constant matrices with appropriate dimensions. Reformulate the model (1) as

$$x(k+1) = \sum_{l=1}^{\hat{n}} m_l(\eta(x)) \left[ \mathbb{A}_l x(k) + \mathbb{B}_l u(k) + \mathbb{B}_{1l} \omega(k) \right],$$
$$z(k) = \sum_{l=1}^{\hat{n}} m_l(\eta(x)) \left[ \mathbb{C}_l x(k) + \mathbb{D}_l u(k) \right], \tag{2}$$

where

$$v_l(\eta(x)) = \prod_{\alpha=1}^{p} \mathcal{V}_\alpha^l(\eta_\alpha(x)) \geq 0,$$
$$m_l(\eta(x)) = \frac{v_l(\eta(x))}{\sum_{l=1}^{\hat{n}} v_l(\eta(x))}, \quad \sum_{l=1}^{\hat{n}} m_l(\eta(x)) = 1.$$

*B. ET Scheme*

In the NCSs, due to the limitation of communication bandwidth, an ET scheme is employed for the reduction of occupation on network resources, which can be described as follows:

$$t_{k+1} = t_k + \min_{j>0} \{ j | \left[ x(t_k + j) - x(t_k) \right]^T \Omega [x(t_k + j)$$
$$- x(t_k)] > \rho x^T(t_k) \Omega x(t_k) \} \ (j \in N), \tag{3}$$

where $\Omega > 0$ is a weighting matrix. $x(t_k)$ is the previously transmitted data. $x(t_k + j)$ denotes the current sampled data. $\rho$ is an appropriate constant that denotes the ET threshold. The data will be transmitted if $x(t_k + j)$ satisfies the ET condition.

Given $\vartheta_k \in \left[0, \bar{\vartheta}\right]$ to be the network-induced delay. Define $\dot{s} = k - \vartheta(k)$. Let $d_m = \min_{j \geq 1} \{ j | \text{T}_k + j \geq \text{T}_{k+1} \}$ ($\text{T}_k = t_k + \vartheta_k$) to decompose $[\text{T}_k, \text{T}_{k+1})$ into $[\text{T}_k, \text{T}_{k+1}) = \cup_{j=1}^{d_m} \theta_j$ , where $\theta_j = \left[ \text{T}_{k+(j-1)}, \text{T}_{k+j} \right)$. $\vartheta(k)$ and $e(k)$ are described as follows:

$$\vartheta(k) = \begin{cases} k - t_k, k \in \theta_1, \\ \vdots \\ k - t_k - j, k \in \theta_{j+1}, \\ \vdots \\ k - t_k - (d_m - 1), k \in \theta_{d_m}, \end{cases} \tag{4}$$

$$e(k) = \begin{cases} 0, k \in \theta_1, \\ \vdots \\ x(t_k) - x(t_k + j), k \in \theta_{j+1}, \\ \vdots \\ x(t_k) - x(t_k + (d_m - 1)), k \in \theta_{d_m}. \end{cases} \tag{5}$$

Form the equations above, it is clear that $\vartheta(k)$ satisfies

$$0 \leq \vartheta_1 \leq \vartheta(k) \leq \bar{\vartheta} + 1 = \vartheta_m,$$

where $\bar{\vartheta} = \max \{\vartheta_k\}$, and $x(t_k) = x(\dot{s}) + e(k)$.

*C. DoS Attacks Compensation Mechanism*

Next, introducing the compensation mechanism [10] to alleviate the unfavorable effect from DoS attacks. The data transmitted to the controller is

$$\widehat{x}(k) = \varpi(k) x(t_k) + (1 - \varpi(k)) \widehat{x}(k - 1), \tag{6}$$

where $\varpi(k)$ is a Bernoulli variable, which shows the randomness of DoS attacks. When $\varpi(k) = 0$, it means that DoS attacks are happening, otherwise $\varpi(k) = 1$.

The mathematical expectation of $\varpi(k)$ is

$$\text{E}\{\varpi(k) = 1\} = \hat{\varpi}, \ \text{E}\{\varpi(k) = 0\} = 1 - \hat{\varpi},$$

where $\hat{\varpi} \in [0, 1)$ is a given constant, and the mathematical variance of Bernoulli variable is $E\{(\varpi(k) - \hat{\varpi})^2\} = \hat{\varpi}(1 - \hat{\varpi}) = \kappa^2$.

*Remark 1:* In this paper, a buffer is used to store the sampling signal that meets the ET condition. When there is no DoS attack occurs, the input signal of the controller is $x(t_k)$. When there is a DoS attack, the input signal of the controller is the previous transmitted signal $\widehat{x}(k - 1)$. This avoids the situation where there is no input signal transmitted to controller when the network is suffering DoS attacks.

*D. Controller Design*

Taking the mismatched membership functions into account, the T-S fuzzy controller with $\hat{n}$ rules is designed as follows:

**Controller Rule** $i$: **IF** $\delta_1(\widehat{x}(k))$ is $\mathcal{L}_1^i$, and $\delta_2(\widehat{x}(k))$ is $\mathcal{L}_2^i$, and ,..., and $\delta_g(\widehat{x}(k))$ is $\mathcal{L}_g^i$, **THEN**

$$u(k) = \mathcal{K}_i \widehat{x}(k), \tag{7}$$

where $\mathcal{L}_\beta^i$ is the fuzzy set ($i = 1, \ldots, \hat{n}$ and $\beta = 1, \ldots, \text{g}$). Define $\delta_\beta(\widehat{x}(k)) = \delta_\beta(\widehat{x})$ to be premise variable. $g$ denotes the number of the premise variables. $\widehat{x}(k) \in \mathbb{R}^x$ is the input vector of the controller. $u(k) \in \mathbb{R}^u$ is the output vector of the controller. $\mathcal{K}_i$ ($i = 1, \ldots, \hat{n}$) are the controller gains. Reformulate the T-S fuzzy controller as

$$u(k) = \sum_{i=1}^{\hat{n}} w_i(\delta(\widehat{x})) \mathcal{K}_i \widehat{x}(k), \tag{8}$$

where

$$\sigma_i(\delta(\widehat{x})) = \prod_{\beta=1}^{g} \mathcal{L}_{\beta}^{i}(\delta_{\beta}(\widehat{x})) \geq 0,$$

$$w_i(\delta(\widehat{x})) = \frac{\sigma_i(\delta(\widehat{x}))}{\sum_{i=1}^{\hat{n}} \sigma_i(\delta(\widehat{x}))}, \quad \sum_{i=1}^{\hat{n}} w_i(\delta(\widehat{x})) = 1.$$

### E. Closed-Loop System (CLS)

Simplify $m_l \triangleq m_l(\eta(x)), w_i \triangleq w_i(\delta(\widehat{x}))$. Combining (2), (3), (6), (8) and $x(t_k) = x(\grave{s}) + e(k)$, the CLS is acquired as

$$\chi(k+1) = \sum_{l=1}^{\hat{n}} \sum_{i=1}^{\hat{n}} m_l w_i [\mathcal{A}_{li1}\chi(k) + (\varpi(k) - \hat{\varpi})$$
$$\times \mathcal{A}_{li2}\chi(k) + \mathcal{A}_{li3}L\chi(\grave{s}) + (\varpi(k)$$
$$- \hat{\varpi})\mathcal{A}_{li4}L\chi(\grave{s}) + \mathcal{A}_{li3}e(k)$$
$$+ (\varpi(k) - \hat{\varpi})\mathcal{A}_{li4}e(k) + \mathcal{B}_{li}\omega(k)],$$

$$z(k) = \sum_{l=1}^{\hat{n}} \sum_{i=1}^{\hat{n}} m_l w_i [\mathcal{C}_{li1}\chi(k) + (\varpi(k) - \hat{\varpi})$$
$$\times \mathcal{C}_{li2}\chi(k) + \mathcal{C}_{li3}L\chi(\grave{s})$$
$$+ (\varpi(k) - \hat{\varpi})\mathcal{C}_{li4}L\chi(\grave{s})$$
$$+ \hat{\varpi}\mathcal{C}_{li3}e(k) + (\varpi(k) - \hat{\varpi})\mathcal{C}_{li4}e(k)], \quad (9)$$

where

$$\chi(k) = [\ x^T(k), \widehat{x}^T(k-1)\ ]^T, L = [\ I\ \ 0\ ],$$

$$\mathcal{A}_{li1} = \begin{bmatrix} \mathbb{A}_l & (1-\hat{\varpi})\mathbb{B}_l\mathcal{K}_i \\ 0 & I(1-\hat{\varpi}) \end{bmatrix}, \mathbb{A}_{li2} = \begin{bmatrix} 0 & -\mathbb{B}_l\mathcal{K}_i \\ 0 & -I \end{bmatrix},$$

$$\mathcal{A}_{li3} = \begin{bmatrix} \hat{\varpi}\mathbb{B}_l\mathcal{K}_i \\ I\hat{\varpi} \end{bmatrix}, \mathcal{A}_{li4} = \begin{bmatrix} \mathbb{B}_l\mathcal{K}_i \\ I \end{bmatrix}, \mathcal{B}_{li} = \begin{bmatrix} \mathbb{B}_{1l} \\ 0 \end{bmatrix},$$

$$\mathcal{C}_{li1} = [\ \mathbb{C}_l\ \ (1-\hat{\varpi})\mathbb{D}_l\mathcal{K}_i\ ], \mathcal{C}_{li2} = [\ 0\ \ -\mathbb{D}_l\mathcal{K}_i\ ],$$

$$\mathcal{C}_{li3} = \hat{\varpi}\mathbb{D}_l\mathcal{K}_i, \mathcal{C}_{li4} = \mathbb{D}_l\mathcal{K}_i.$$

## III. MAIN RESULTS

### A. Performance Analysis

In order to assure system (9) under $H_{\infty}$ performance is asymptotically stable, sufficient conditions are attained from the following theorem.

*Theorem 1:* Given constants $\rho > 0, \vartheta_m > 0, \hat{\varpi} > 0$ and $\gamma > 0$, if the MFs satisfy $w_i - \varepsilon_i m_i \geq 0$ $(0 < \varepsilon_i \leq 1)$, and there exist matrices $P > 0, Q_1 > 0, Q_2 > 0, \Omega > 0, \Delta_l = \Delta_l^T$ and $R$ satisfying $(l, i = 1, 2, ..., \hat{n})$:

$$\begin{bmatrix} Q_2 & R \\ * & Q_2 \end{bmatrix} \geq 0, \quad (10)$$

$$\Psi_{li} - \Delta_l < 0, \quad (11)$$

$$\varepsilon_l \Psi_{ll} - \varepsilon_l \Delta_l + \Delta_l < 0, \quad (12)$$

$$\varepsilon_i \Psi_{li} - \varepsilon_i \Delta_l + \Delta_l + \varepsilon_l \Psi_{il} - \varepsilon_l \Delta_i + \Delta_i < 0, l < i, \quad (13)$$

where

$$\Psi_{li} = \begin{bmatrix} \Theta_1 & \Theta_{1li} & \Theta_{2li} & \Theta_{3li} & \Theta_{4li} & \Theta_{5li} & \Theta_{6li} \\ \star & -P & 0 & 0 & 0 & 0 & 0 \\ \star & \star & -P & 0 & 0 & 0 & 0 \\ \star & \star & \star & -Q_2 & 0 & 0 & 0 \\ \star & \star & \star & \star & -Q_2 & 0 & 0 \\ \star & \star & \star & \star & \star & -I & 0 \\ \star & \star & \star & \star & \star & \star & -I \end{bmatrix},$$

$$\Theta_1 = \begin{bmatrix} \Xi_1 & \Xi_2 & -L^T R & 0 & 0 \\ \star & \Xi_3 & Q_2+R & \rho\Omega & 0 \\ \star & \star & -Q_1-Q_2 & 0 & 0 \\ \star & \star & \star & (\rho-1)\Omega & 0 \\ \star & \star & \star & \star & -\gamma^2 I \end{bmatrix},$$

$$\Xi_1 = -P - L^T Q_2 L + L^T Q_1 L,$$
$$\Xi_2 = L^T Q_2 + L^T R,$$
$$\Xi_3 = -2Q_2 - R + \rho\Omega,$$
$$\Theta_{1li} = [\ P\mathcal{A}_{li1}\ \ P\mathcal{A}_{li3}\ \ 0\ \ P\mathcal{A}_{li3}\ \ P\mathcal{B}_{li}\ ]^T,$$
$$\Theta_{2li} = \kappa\ [\ P\mathcal{A}_{li2}\ \ P\mathcal{A}_{li4}\ \ 0\ \ P\mathcal{A}_{li4}\ \ 0\ ]^T,$$
$$\Theta_{3li} = \vartheta_m\ [\ \mathcal{A}_{li1} - I\ \ \mathcal{A}_{li3}\ \ 0\ \ \mathcal{A}_{li3}\ \ \mathcal{B}_{li}\ ]^T,$$
$$\Theta_{4li} = \kappa\ [\ \mathcal{A}_{li2}\ \ \mathcal{A}_{li4}\ \ 0\ \ \mathcal{A}_{li4}\ \ 0\ ]^T,$$
$$\Theta_{5li} = [\ \mathcal{C}_{li1}\ \ \mathcal{C}_{li3}\ \ 0\ \ \mathcal{C}_{li3}\ \ 0\ ]^T,$$
$$\Theta_{6li} = [\ \mathcal{C}_{li2}\ \ \mathcal{C}_{li3}\ \ 0\ \ \mathcal{C}_{li4}\ \ 0\ ]^T,$$

then, the asymptotic stability of system (9) under $H_{\infty}$ performance is obtained.

*Proof 1:* Firstly, select the following Lyapunov-Krasovskii functional

$$\mathcal{V}(k) = \mathcal{V}_1(k) + \mathcal{V}_2(k) + \mathcal{V}_3(k), \quad (14)$$

where

$$\mathcal{V}_1(k) = \chi^T(k)P\chi(k),$$

$$\mathcal{V}_2(k) = \sum_{\grave{\alpha}=k-\vartheta_m}^{k-1} \chi^T(\grave{\alpha})L^T Q_1 L\chi(\grave{\alpha}),$$

$$\mathcal{V}_3(k) = \sum_{\grave{\alpha}=-\vartheta_m}^{-1} \sum_{\grave{\beta}=k+\grave{\alpha}}^{k-1} [\chi(\grave{\beta}+1) - \chi(\grave{\beta})]^T L^T Q_2$$
$$\times L[\chi(\grave{\beta}+1) - \chi(\grave{\beta})].$$

Simplify $\grave{s}_m = k - \vartheta_m$. The difference of $\mathcal{V}(k)$ is calculated as

$$\Delta\mathcal{V}(k) = \Delta\mathcal{V}_1(k) + \Delta\mathcal{V}_2(k) + \Delta\mathcal{V}_3(k),$$

where

$$E\{\Delta\mathcal{V}_1(k)\} = \chi^T(k+1)P\chi(k+1) - \chi^T(k)P\chi(k),$$

$$E\{\Delta \mathcal{V}_2(k)\} = \chi^T(k) L^T Q_1 L \chi(k)$$
$$- \chi^T(\grave{s}_m) L^T Q_1 L \chi(\grave{s}_m),$$
$$E\{\Delta \mathcal{V}_3(k)\} = \vartheta_m^{\ 2} [\chi(k+1) - \chi(k)]^T L^T Q_2 L [\chi(k+1)$$
$$- \chi(k)] - \vartheta_m \sum_{\grave{\alpha}=\grave{s}_m}^{k-1} [\chi(\grave{\alpha}+1) - \chi(\grave{\alpha})]^T L^T$$
$$\times Q_2 L [\chi(\grave{\alpha}+1) - \chi(\grave{\alpha})].$$

Since $\begin{bmatrix} Q_2 & R \\ * & Q_2 \end{bmatrix} \geq 0$, based on Jensen inequality, $E\{\Delta \mathcal{V}_3(k)\}$ can be obtained

$$E\{\Delta \mathcal{V}_3(k)\} = \vartheta_m^{\ 2} [\chi(k+1) - \chi(k)]^T L^T Q_2 L [\chi(k+1)$$
$$- \chi(k)] - \vartheta_m \sum_{\grave{\alpha}=\grave{s}_m}^{k-1} [\chi(\grave{\alpha}+1) - \chi(\grave{\alpha})]^T L^T$$
$$\times Q_2 L [\chi(\grave{\alpha}+1) - \chi(\grave{\alpha})]$$
$$\leq \vartheta_m^2 [\chi(k+1) - \chi(k)]^T L^T Q_2 L [\chi(k+1)$$
$$- \chi(k)] - [\chi(k) - \chi(\grave{s})]^T L^T Q_2 L [\chi(k)$$
$$- \chi(\grave{s})] - [\chi(\grave{s}) - \chi(\grave{s}_m)]^T L^T Q_2 L [\chi(\grave{s})$$
$$- \chi(\grave{s}_m)] + [\chi(k) - \chi(\grave{s})]^T L^T R L [\chi(\grave{s})$$
$$- \chi(\grave{s}_m)] + [\chi(\grave{s}) - \chi(\grave{s}_m)]^T L^T R L$$
$$\times [\chi(k) - \chi(\grave{s})]. \tag{15}$$

Defining $\varrho(k) = \left[\chi^T(k), \chi^T(k - \vartheta(k)) L^T, \chi^T(\grave{s}_m) L^T, e^T(k), \omega^T(k)\right]^T$, and taking ET condition into consideration, we have

$$E\{\Delta \mathcal{V}(k) + z^T(k) z(k) - \gamma^2 \omega^T(k) \omega(k)\}$$
$$\leq \sum_{l=1}^{\hat{n}} \sum_{i=1}^{\hat{n}} m_l w_i \varrho^T(k) \Psi_{li} \varrho(k). \tag{16}$$

Applying Schur complement for (16), the following inequality can be attained

$$\Delta \mathcal{V}(k) - \gamma^2 \omega^T(k) \omega(k) + z^T(k) z(k) < 0. \tag{17}$$

It is clear that

$$\sum_{k=1}^{\infty} \Delta \mathcal{V}(k) < \sum_{k=1}^{\infty} \gamma^2 \omega^T(k) \omega(k) - \sum_{k=1}^{\infty} z^T(k) z(k),$$

then we have

$$\mathcal{V}(k) < \sum_{k=1}^{\infty} \gamma^2 \omega^T(k) \omega(k) - \sum_{k=1}^{\infty} z^T(k) z(k). \tag{18}$$

Since $\mathcal{V}(k) > 0$, we can acquire $\sum_{k=1}^{\infty} \gamma^2 \omega^T(k) \omega(k) > \sum_{k=1}^{\infty} z^T(k) z(k)$, the condition of $H_\infty$ performance for CLS (9) is proofed. As can be seen in (18), when $\omega(k) = 0$, it can be obtained that $\Delta \mathcal{V}(k) < 0$, and the CLS (9) is asymptotically stable.

Next, introducing a slack matrix $\Delta_l$ to reduce the conservativeness of the obtained results. Taking

$\sum_{l=1}^{\hat{n}} \sum_{i=1}^{\hat{n}} m_l (m_i - w_i) \Delta_l = 0$ into consideration, it is easy to acquire

$$\sum_{l=1}^{\hat{n}} \sum_{i=1}^{\hat{n}} m_l w_i \Psi_{li}$$
$$= \sum_{l=1}^{\hat{n}} \sum_{i=1}^{\hat{n}} m_l (m_i - w_i + \varepsilon_i m_i - \varepsilon_i m_i) \Delta_l + \sum_{l=1}^{\hat{n}} \sum_{i=1}^{\hat{n}} m_l w_i \Psi_{li}$$
$$= \sum_{l=1}^{\hat{n}} \sum_{i=1}^{\hat{n}} m_l m_i (\varepsilon_i \Psi_{li} - \varepsilon_i \Delta_l + \Delta_l)$$
$$+ \sum_{l=1}^{\hat{n}} \sum_{i=1}^{\hat{n}} m_l (w_i - \varepsilon_i m_i) (\Psi_{li} - \Delta_l)$$
$$= \sum_{l=1}^{\hat{n}} m_l^2 (\varepsilon_l \Psi_{ll} - \varepsilon_l \Delta_l + \Delta_l)$$
$$+ \sum_{l=1}^{\hat{n}-1} \sum_{i=1}^{\hat{n}} m_l m_i (\varepsilon_i \Psi_{li} - \varepsilon_i \Delta_l + \Delta_l + \varepsilon_l \Psi_{il} - \varepsilon_l \Delta_i + \Delta_i)$$
$$+ \sum_{l=1}^{\hat{n}} \sum_{i=1}^{\hat{n}} m_l (w_l - \varepsilon_i m_i) (\Psi_{li} - \Delta_l).$$

Based on $w_i - \varepsilon_i m_i \geq 0$ $(0 < \varepsilon_i \leq 1)$ and (11)-(13), $E\{\Delta \mathcal{V}(k) + z^T(k) z(k) - \gamma^2 \omega^T(k) \omega(k)\} \leq 0$ is attained. The proof is completed.

*B. Controller Design*

Based on Theorem 1, this subsection focuses on determining the appropriate gains of the controller.

*Theorem 2:* Given constants $\rho > 0, \vartheta_m > 0, \hat{\varpi} > 0$ and $\gamma > 0$, if the MFs satisfy $w_i - \varepsilon_i m_i \geq 0$ $(0 < \varepsilon_i \leq 1)$, and there exist matrices $\mathcal{X} > 0, \tilde{Q}_1 > 0, \tilde{Q}_2 > 0, \tilde{\Omega} > 0, \tilde{\Delta}_l = \tilde{\Delta}_l^T$ and $\tilde{R}$ satisfying $(l, i = 1, 2, ..., \hat{n})$:

$$\begin{bmatrix} \tilde{Q}_2 & \tilde{R} \\ * & \tilde{Q}_2 \end{bmatrix} \geq 0, \tag{19}$$
$$\tilde{\Psi}_{li} - \tilde{\Delta}_l < 0, \tag{20}$$
$$\varepsilon_l \tilde{\Psi}_{ll} - \varepsilon_l \tilde{\Delta}_l + \tilde{\Delta}_l < 0, \tag{21}$$
$$\varepsilon_i \tilde{\Psi}_{li} - \varepsilon_i \tilde{\Delta}_l + \tilde{\Delta}_l + \varepsilon_l \tilde{\Psi}_{il} - \varepsilon_l \tilde{\Delta}_i + \tilde{\Delta}_i < 0, l < i, \tag{22}$$

where

$$\Psi_{li} = \begin{bmatrix} \widehat{\Theta}_1 & \widehat{\Theta}_{1li} & \widehat{\Theta}_{2li} & \tilde{\Theta}_{3li} & \widehat{\Theta}_{4li} & \widehat{\Theta}_{5li} & \widehat{\Theta}_{6li} \\ \star & \widehat{\Theta}_{Pli} & 0 & 0 & 0 & 0 & 0 \\ \star & \star & \widehat{\Theta}_{Pli} & 0 & 0 & 0 & 0 \\ \star & \star & \star & \widehat{\Theta}_{Qli} & 0 & 0 & 0 \\ \star & \star & \star & \star & \widehat{\Theta}_{Qli} & 0 & 0 \\ \star & \star & \star & \star & \star & -I & 0 \\ \star & \star & \star & \star & \star & \star & -I \end{bmatrix},$$

$$\widehat{\Theta}_1 = \begin{bmatrix} \widehat{\Xi}_{11} & \widehat{\Xi}_{12} & \widehat{\Xi}_{13} & 0 \\ \star & \widehat{\Xi}_{22} & \widehat{\Xi}_{23} & \widehat{\Xi}_{24} \\ \star & \star & \widehat{\Xi}_{33} & 0 \\ \star & \star & \star & \widehat{\Xi}_{44} \end{bmatrix},$$

$$\widehat{\Xi}_{11} = \begin{bmatrix} -\mathcal{X} - \tilde{Q}_2 + \tilde{Q}_1 & 0 \\ 0 & -\mathcal{X}^T \end{bmatrix},$$

$$\widehat{\Xi}_{12} = \begin{bmatrix} \tilde{Q}_2 + \tilde{R} \\ 0 \end{bmatrix}, \widehat{\Xi}_{22} = \begin{bmatrix} -2\tilde{Q}_2 - \tilde{R} + \rho\tilde{\Omega} \end{bmatrix},$$

$$\widehat{\Xi}_{13} = \begin{bmatrix} -\tilde{R} \\ 0 \end{bmatrix}, \widehat{\Xi}_{23} = \begin{bmatrix} \tilde{Q}_2 + \tilde{R} \end{bmatrix}, \widehat{\Xi}_{33} = \begin{bmatrix} \tilde{Q}_1 - \tilde{Q}_2 \end{bmatrix},$$

$$\widehat{\Xi}_{24} = \begin{bmatrix} \rho\tilde{\Omega} & 0 \end{bmatrix}, \widehat{\Xi}_{44} = \begin{bmatrix} (\rho-1)\tilde{\Omega} & 0 \\ 0 & -\gamma^2 I \end{bmatrix},$$

$$\widehat{\Theta}_{Pli} = \begin{bmatrix} -\mathcal{X}^T & 0 \\ * & -\mathcal{X}^T \end{bmatrix}, \widehat{\Theta}_{Qli} = \begin{bmatrix} 2\mathcal{X}^T - \tilde{Q}_2^T \end{bmatrix},$$

$$\widehat{\Theta}_{1li} = \begin{bmatrix} \widehat{\Theta}_{11li} & \widehat{\Theta}_{12li} \end{bmatrix}^T,$$

$$\widehat{\Theta}_{11li} = \begin{bmatrix} \mathbb{A}_l \mathcal{X} & (1-\hat{\varpi})\mathbb{B}_l\tilde{\mathcal{K}}_i & \hat{\varpi}\mathbb{B}_l\tilde{\mathcal{K}}_i \\ 0 & (1-\hat{\varpi})\mathcal{X} & \hat{\varpi}\mathcal{X} \end{bmatrix},$$

$$\widehat{\Theta}_{12li} = \begin{bmatrix} 0 & \hat{\varpi}\mathbb{B}_l\tilde{\mathcal{K}}_i & \mathbb{B}_{1l} \\ 0 & \hat{\varpi}\mathcal{X} & 0 \end{bmatrix}, \widehat{\Theta}_{2li} = \begin{bmatrix} \widehat{\Theta}_{21li} & \widehat{\Theta}_{22li} \end{bmatrix}^T,$$

$$\widehat{\Theta}_{21li} = \begin{bmatrix} 0 & -\kappa\mathbb{B}_l\tilde{\mathcal{K}}_i & \kappa\mathbb{B}_l\tilde{\mathcal{K}}_i \\ 0 & -\kappa\mathcal{X} & \kappa\mathcal{X} \end{bmatrix},$$

$$\widehat{\Theta}_{22li} = \begin{bmatrix} 0 & \kappa\mathbb{B}_l\tilde{\mathcal{K}}_i & 0 \\ 0 & \kappa\mathcal{X} & 0 \end{bmatrix}, \widehat{\Theta}_{3li} = \begin{bmatrix} \widehat{\Theta}_{31li} & \widehat{\Theta}_{32li} \end{bmatrix}^T,$$

$$\widehat{\Theta}_{31li} = \begin{bmatrix} \vartheta_m(\mathbb{A}_l - I)\mathcal{X} & \vartheta_m(1-\hat{\varpi})\mathbb{B}_l\tilde{\mathcal{K}}_i \end{bmatrix},$$

$$\widehat{\Theta}_{32li} = \begin{bmatrix} \vartheta_m\kappa\mathbb{B}_l\tilde{\mathcal{K}}_i & 0 & \vartheta_m\hat{\varpi}\mathbb{B}_l\tilde{\mathcal{K}}_i & 0 \end{bmatrix},$$

$$\widehat{\Theta}_{4li} = \begin{bmatrix} \widehat{\Theta}_{41li} & \widehat{\Theta}_{42li} \end{bmatrix}^T,$$

$$\widehat{\Theta}_{41li} = \begin{bmatrix} 0 & -\vartheta_m\kappa\mathbb{B}_l\tilde{\mathcal{K}}_i & \vartheta_m\kappa\mathbb{B}_l\tilde{\mathcal{K}}_i \end{bmatrix},$$

$$\widehat{\Theta}_{42li} = \begin{bmatrix} 0 & \vartheta_m\kappa\mathbb{B}_l\tilde{\mathcal{K}}_i & 0 \end{bmatrix}, \widehat{\Theta}_{5li} = \begin{bmatrix} \widehat{\Theta}_{51li} & \widehat{\Theta}_{52li} \end{bmatrix}^T,$$

$$\widehat{\Theta}_{51li} = \begin{bmatrix} \mathbb{C}_l\mathcal{X} & \hat{\varpi}(1-\hat{\varpi})\mathbb{D}_l\tilde{\mathcal{K}}_i & \hat{\varpi}\mathbb{D}_l\tilde{\mathcal{K}}_i \end{bmatrix}$$

$$\widehat{\Theta}_{52li} = \begin{bmatrix} 0 & \hat{\varpi}\mathbb{D}_l\tilde{\mathcal{K}}_i & 0 \end{bmatrix}, \widehat{\Theta}_{6li} = \begin{bmatrix} \widehat{\Theta}_{61li} & \widehat{\Theta}_{62li} \end{bmatrix}^T,$$

$$\widehat{\Theta}_{61li} = \begin{bmatrix} 0 & -\kappa\mathbb{D}_l\tilde{\mathcal{K}}_i & \kappa\mathbb{D}_l\tilde{\mathcal{K}}_i \end{bmatrix},$$

$$\widehat{\Theta}_{62li} = \begin{bmatrix} 0 & \kappa\mathbb{D}_l\tilde{\mathcal{K}}_i & 0 \end{bmatrix},$$

then, system (9) under $H_\infty$ performance to be asymptotically stable is obtained. The controller gains are acquired as

$$\mathcal{K}_i = \tilde{\mathcal{K}}_i \mathcal{X}^{-1}.$$

*Proof 2:* Define:

$$P = \text{diag}\{P_1, P_1\}, \mathcal{X} = P_1^{-1}.$$

Pre and post multiplying (11)-(13) by $P_2 = \text{diag}\{\mathcal{X}, \mathcal{X}, \mathcal{X}, \mathcal{X}, \mathcal{X}, I, \mathcal{X}, \mathcal{X}, \mathcal{X}, \mathcal{X}, Q_2^{-1}, Q_2^{-1}, I, I\}$ and $P_2^T$, respectively.

Define:

$$\tilde{Q}_1 = \mathcal{X}^T Q_1 \mathcal{X}, \ \tilde{Q}_2 = \mathcal{X}^T Q_2 \mathcal{X},$$
$$\tilde{R} = \mathcal{X}^T R \mathcal{X}, \ \tilde{\Omega} = \mathcal{X}^T \Omega \mathcal{X}, \ \tilde{\mathcal{K}}_i = \mathcal{K}_i \mathcal{X}.$$

Then, (20)-(22) can be acquired. The proof is completed.

## IV. SIMULATION VERIFICATION

In this section, we provide a numerical simulation example to validate the effectiveness of the proposed control mechanism, the concrete content is illustrated by the following example.

Considering the following T-S fuzzy model

$$\mathbb{A}_1 = \begin{bmatrix} -0.0486 & 0.4262 \\ -1.6264 & -0.2168 \end{bmatrix}, \mathbb{B}_1 = \begin{bmatrix} 0.3616 \\ 0.5608 \end{bmatrix},$$

$$\mathbb{A}_2 = \begin{bmatrix} -0.0486 & 0.4262 \\ -1.6264 & -0.2168 \end{bmatrix}, \mathbb{B}_2 = \begin{bmatrix} 0.3616 \\ 0.5608 \end{bmatrix},$$

$$\mathbb{B}_{11} = \begin{bmatrix} 0.1 \\ 0.5 \end{bmatrix}, \mathbb{C}_1 = \begin{bmatrix} 0.35 & -0.6 \end{bmatrix}, \mathbb{D}_1 = 0.05,$$

$$\mathbb{B}_{12} = \begin{bmatrix} 0 \\ -1.15 \end{bmatrix}, \mathbb{C}_2 = \begin{bmatrix} 0.35 & -0.6 \end{bmatrix}, \mathbb{D}_2 = -0.02.$$

Assuming $x_1(k) \in [-2, 2]$ and the membership functions are defined as

$$m_1(x_1(k)) = \frac{4 - x_1^2(k)}{4}, \ m_2(x_1(k)) = 1 - m_1(x_1(k)).$$

Next, define the membership functions of the controller as $w_1(\hat{x}_1(k)) = (1 - \sin^2(\hat{x}_1(k)))/2$ and $w_2(\hat{x}_1(k)) = 1 - w_1(\hat{x}_1(k))$. Select $\varepsilon_1 = 0.5$, $\varepsilon_2 = 0.7$, $\rho = 0.31$, $\hat{\varpi} = 0.5$ and $\vartheta_m = 0.1$, and the controller gains are acquired as

$$\mathcal{K}_1 = \begin{bmatrix} 0.2166 & -0.0026 \end{bmatrix},$$
$$\mathcal{K}_2 = \begin{bmatrix} 0.2525 & 0.0579 \end{bmatrix}.$$

Suppose the initial state is $x(0) = \begin{bmatrix} 0.5, -0.2 \end{bmatrix}^T$, and the sampling period $h = 0.2s$. The disturbance is assumed as
$$w(k) = \begin{cases} -1.5\cos(1.2k)e^{-0.1k}, 0 \le k \le 14.4 \\ 0, \quad\quad\quad\quad\quad\quad\quad\quad \text{else} \end{cases}.$$

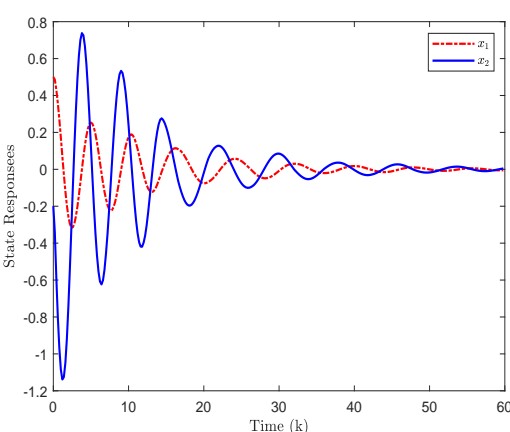

Fig. 1. State responses.

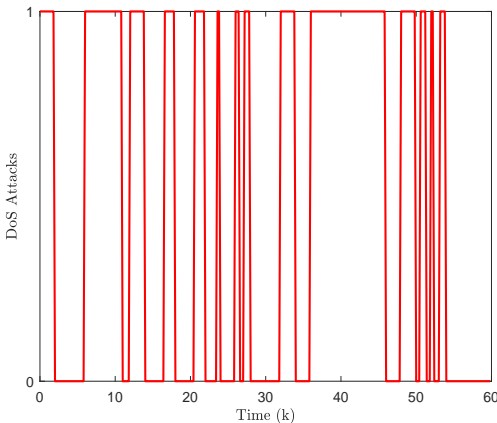

Fig. 2. The occurrence instants of DoS attacks.

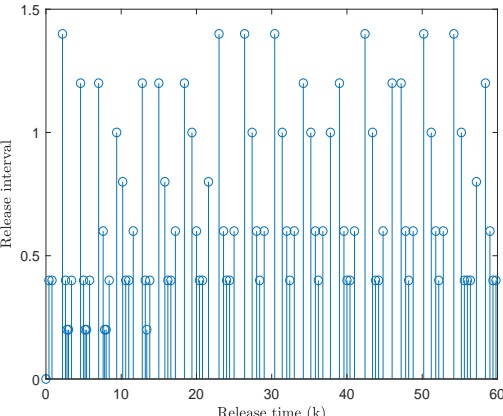

Fig. 3. Release instants and intervals of ET scheme.

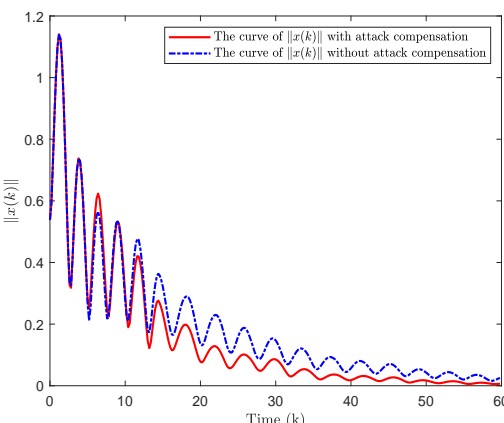

Fig. 4. Comparision of attack compensation.

To ensure the successful transmission of the initial state, assuming that no DoS attack happens when $k \in [0, 2)$. Fig. 1 illustrates the responses of $x_1(k)$ and $x_2(k)$. The occurrence

interval of DoS attacks is illustrated in Fig. 2. The triggering instants and intervals of the ET scheme is shown in Fig. 3. In addition, to demonstrate effectiveness of attack compensation mechanism, a comparative simulation result is illustrated in Fig. 4. From Fig. 4, it is proved that the application of attack compensation mechanism can effectively reduce the unfavorable impact of DoS attacks, making the system stabilize faster.

## V. CONCLUSION

In this paper, we have examined the design problem of ET-based T-S fuzzy controller subject to malicious attacks for networked control systems. An ET scheme has been utilized to save network resources. By taking the DoS attacks compensation mechanism, the unfavorable effects of DoS attacks have been effectively reduced. Then, the Lyapunov stability theory has been utilized to identify the sufficient condition for asymptotic stability of the CLS. Eventually, simulation verification have been provided to testify the effectiveness of the proposed control strategy.

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
