# OpenReview forum: "Event-Triggered-Based Fuzzy Control for Networked Control Systems with Compensation Mechanism against DoS Attacks"
_IEEE.org/ICIST/2024/Conference — IEEE ICIST 2024 Conference Submission_

### Official Review · Reviewer_7Rs3 · 2024-08-21
**Accept**

**Rating:** 7
**Confidence:** 5

**Review:**

This manuscript primarily investigates the design problem of event-triggered fuzzy controllers for networked control systems subject to denial-of-service (DoS) attacks. It proposes an event-triggered scheme to save network resources and develops a DoS attack compensation scheme. Overall, the manuscript is well-written, well-organized, with clear derivations, and the simulation results support the research findings. However, the authors should address some formatting issues, such as the use of block paragraphs in certain sections.

---

### Official Review · Reviewer_3ANM · 2024-08-22
**Event-Triggered-Based Fuzzy Control for Networked Control Systems with Compensation Mechanism against DoS Attacks**

**Rating:** 7
**Confidence:** 5

**Review:**

This paper designed an event-triggered fuzzy controller for networked control systems (NCSs) that can mitigate the effects of denial-of-service (DoS) attacks while conserving communication resources. This topic is interesting, the following comments need to further consider: (1) The abstract contains a few grammatical issues and could benefit from clearer language in certain parts. For example, the phrase "an ET scheme is introduced to save network resources and then a DoS attack compensation mechanism is applied in the ET to controller channel to maintain the transmission of the data" is somewhat convoluted and could be rephrased for better clarity. (2) Discussion of the related work on other approaches should be extended. (3) The contributions should be illustrated in a clearer manner. For example, what is the main improvement of the paper compared to the existing results. (4) The format of the references should be unified.

---

### Official Review · Reviewer_yxPu · 2024-08-22
**Manuscript Accept**

**Rating:** 7
**Confidence:** 5

**Review:**

Introduce the advantage of the proposed event-triggered scheme.
By applying the attack compensation mechanism, does the efficacy of data transmission return to the situation before the system suffered from DoS attacks?
How do the authors get the moments of occurring DoS attacks?

---

### Decision · Program_Chairs · 2024-09-08

Accept (Oral)